Genome-wide characterization and expression analysis of GRAS gene family in pepper (Capsicum annuum L.)

Liu Baoling
http://orcid.org/0000-0002-5797-4880 Sun Yan
Xue Jinai
Jia Xiaoyun
Li Runzhi rli2001@126.com
Institute of Molecular Agriculture and Bioenergy, Shanxi Agricultural University , Jinzhong City , China
Hesham Abd El-Latif
Electronic publication date: 2018 May 29
Publication date: 2018
Volume: 6
Electronic Location ID: e4796
Received 2018 Jan 10; Accepted 2018 Apr 29
Copyright: © 2018 Liu et al.
Copyright year: 2018
Copyright holder: Liu et al.
License: This is an open access article distributed under the terms of the Creative Commons Attribution License, which permits unrestricted use, distribution, reproduction and adaptation in any medium and for any purpose provided that it is properly attributed. For attribution, the original author(s), title, publication source (PeerJ) and either DOI or URL of the article must be cited.
License URL: https://creativecommons.org/licenses/by/4.0/

Keywords: GRAS genes, Gene expression, Abiotic stress, Duplication, Pepper

Funding: National Natural Science Foundation of China 30971806, 31201266, and 31401430 State Ministry of Agriculture “948” Project 2014-Z39 Shanxi Province Key Projects of Coal-based Science and Technology FT-2014-01 Shanxi Scholarship Council of China 2015-064 Key Project of the Key Research and Development Program of Shanxi Province, China 201603D312005 This work was supported by the National Natural Science Foundation of China (Grant No. 30971806, 31201266, and 31401430), the State Ministry of Agriculture “948” Project (2014-Z39), Shanxi Province Key Projects of Coal-based Science and Technology (FT-2014-01), Research Project Supported by Shanxi Scholarship Council of China (2015-064), and the Key Project of the Key Research and Development Program of Shanxi Province, China (Grant No. 201603D312005). The funders had no role in study design, data collection and analysis, decision to publish, or preparation of the manuscript.

==============================
Plant-specific GRAS transcription factors regulate various biological processes in plant growth, development and stress responses. However, this important gene family was not fully characterized in pepper (Capsicum annuum L.), an economically important vegetable crop. Here, a total of 50 CaGRAS members were identified in pepper genome and renamed by their respective chromosomal distribution. Genomic organization revealed that most CaGRAS genes (84%) have no intron. Phylogenetic analysis divided pepper CaGRAS members into 10 subfamilies, with each having distinct conserved domains and functions. For the expansion of the GRAS genes in pepper, segmental duplication contributed more than tandem duplication did. Gene expression analysis in various tissues demonstrated that most of CaGRAS genes exhibited a tissue- and development stage-specific expression pattern, uncovering their potential functions in pepper growth and development. Moreover, 21 CaGRAS genes were differentially expressed under cold, drought, salt and gibberellin acid (GA) treatments, indicating that they may implicated in plant response to abiotic stress. Notably, GA responsive cis-elements were detected in the promoter regions of the majority of CaGRAS genes, suggesting that CaGRAS may involve in signal cross-talking. The first comprehensive analysis of GRAS gene family in pepper genome by this study provide insights into understanding the GRAS-mediated regulation network, benefiting the genetic improvements in pepper and some other relative plants.

Introduction

GRAS proteins, a group of plant-specific transcription regulators, are named after the acronyms of three initially identified members: GAI, RGA and SCR. Typically, GRAS proteins are composed of 400–770 amino acid residues (Bolle, 2004; Pysh et al., 1999), and contain several highly-conserved motifs at their C-termini but great variation in length and sequence at their N-termini. The consecutive conserved motifs at C-terminal region include LHR I, VHIID, LHR II, PFYRE and SAW (Pysh et al., 1999; Sun et al., 2011), which contribute to protein function. The structure of VHIID with its flanking two leucine heptad repeats (LHR I and LHR II) is critical for protein–protein interaction. The mutagenesis of PFYRE and SAW motifs displayed distinct phenotype abnormality in Arabidopsis thaliana, indicating that they may contribute to the structural integrity of GRAS family (Wang et al., 2014; Itoh et al., 2002; Silverstone, Ciampaglio & Sun, 1998). Except for two conserved N-terminal motifs (DELLA and TVHYNP) characterized only for the members of DELLA subgroup, N-termini of GRAS proteins display large divergence, which may determine functional specificity of such regulatory proteins (Sun et al., 2011).

Recently, GRAS genes have been characterized in a number of plant species, such as A. thaliana, rice (Oryza sativa), tomato (Solanum lycopersicum), poplar (Populus trichocarpa), Chinese cabbage (Brassica rapa ssp. pekinensis), maize (Zea mays), Medicago truncatula and pine (Pinus radiata) (Abarca et al., 2014; Huang et al., 2015; Lu et al., 2015; Ma et al., 2010; Song et al., 2014; Tian et al., 2004; Zhang et al., 2017). According to the conserved motifs and sequence similarity, GRAS family members in two model plants, Arabidopsis and rice, were classified into eight distinct subfamilies, namely DELLA, HAM, LISCL, PAT1, LAS, SCR, SHR and SCL3 (Tian et al., 2004). However, the number of subfamily was ranged from eight to 16 in other plants such as Prunus mume, tomato and maize, suggesting that species-specific subfamily may exist in those plants unexamined yet.

The known studies demonstrated that GRAS proteins function in various physiological processes during plant growth and development, including axillary meristem formation, root development, gametogenesis, phytochrome and gibberellin acid (GA) signal transduction and the response to stresses (Cenci & Rouard, 2017). Considering the fact that amino acid sequences in each subfamily are highly homologous, each group might possess distinct functions. For example, SCR and SHR, are both found to regulate root and shoot radial organization via a SCR/SHR complex in Arabidopsis (Cui et al., 2007; Helariutta et al., 2000). DELLA members usually act as the inhibitors of GA signaling perception (Sun & Gubler, 2004). SCL3 mainly expressed in endodermis is essential for integrating downstream pathways of SCR/SHR and GA/DELLA, and controlling GA homeostasis during root development (Zhang et al., 2011). AtSCL13 from the PAT1 subfamily participate in phytochrome-B (phyB) signal transduction (Bolle, Koncz & Chua, 2000), whereas other members (PAT1, SCL5 and SCL21) from the same subfamily mainly function as positive regulators mediating phyA signaling pathway to control plant development (Torres-Galea et al., 2006). OsMOC1, a putative GRAS protein, has been proven as a positive regulator of rice tillering, important in the direct control of grain yield (Li et al., 2003). Although these GRAS members were functionally characterized in model plants, large amount of GRAS proteins remain to be elucidated for their functions, particularly in agricultural plants.

Pepper (Capsicum annuum L.) is an economically important vegetable. It has tremendous value for providing food, spice, coloring agent, pharmaceuticals and ornamental products (Kim et al., 2014; Qin et al., 2014). In 2013, the total pepper production in the world already reached 34.9 million tons, making it the second largest Solanaceae crop after tomato (Kim et al., 2014). The accomplishment of whole genome sequencing in 2014 provides a platform for us to conduct genome-wide analysis for an entire gene family and to explore the right gene which is critical for pepper growth and development (Kim et al., 2014; Qin et al., 2014). By far, transcription factor families, such as WRKY, Dof, SBP-Box and Hsp70 have been characterized in pepper (Guo et al., 2015b, 2016; Wu et al., 2016). However, pepper GRAS proteins and their functional specificity have not been characterized in detail. Here, we firstly describe the entire members of GRAS family in pepper using comparative genomic tools and experimental verification. A total of 50 CaGRAS genes were identified from pepper genome. The intron/exon organization and protein structure of each GRAS member were also characterized, together with their phylogenetic relationships and chromosomal locations. Subsequently, we examined the function diversity of CaGRAS members by conserved motif analysis, followed by real-time PCR to profile their expression patterns in different tissues and various stress treatments. The present study provides essential knowledge to further illuminate molecular functions of GRAS genes in regulation of pepper growth and development as well as environmental responses.

Materials and Methods

Identification and annotation of pepper GRAS genes

Whole genome data for pepper cv. CM334 and cv. Zunla-1 were used for this study, and their genomic information were downloaded from http://peppergenome.snu.ac.kr/download.php and http://peppersequence.genomics.cn/, respectively (Kim et al., 2014; Qin et al., 2014). Arabidopsis GRAS genes were obtained from TAIR (https://www.Arabidopsis.org/), whereas rice GRAS genes were downloaded from RGAP (http://rice.plantbiology.msu.edu) (Tian et al., 2004). The tomato GRAS information was obtained from SGN (https://solgenomics.net/) (Niu et al., 2017). The latest Hidden Markov Model (HMM) of GRAS domain (PF03514.11) (http://pfam.sanger.ac.uk/) was used as a BLAST query to search against the entire protein datasets of cv. CM334 and cv. Zunla-1 with an E-value of 1e−5 using HMMER 3.0 (Huang et al., 2015). Meanwhile, all AtGRAS and OsGRAS proteins were used as queries to search against the two pepper databases using default parameters. The length of the hit out of the range from 350 to 820 aa was rejected. In order to validate their putative accuracy, conserved domains essential for GRAS proteins were evaluated by SMART (http://smart.embl-heidelberg.de/) and PFAM database. Finally, all outputs from two independent databases were aligned and those having similar GRAS core domain were deemed as the same gene. After these stringent criterions, sequences with the presence of GRAS domain were retained for further analysis. In our study, we refer to the variety cv. CM334 as the reference for subsequent whole genome-wide analysis.

Phylogenetic analysis of CaGRAS genes

All screened GRAS proteins from pepper, Arabidopsis, rice and tomato were used for multiple alignments by ClustalW program (Larkin et al., 2007). Gene IDs of GRAS members used in this study were listed in Table S1. Arabidopsis and rice are most common used model plants for researching genetic correlations, and tomato is another model plant of the Solanaceae family, which is closely related to pepper. Then maximum likelihood method was adopted to generate an unrooted phylogenetic tree based on alignment results. Reliability of phylogenetic tree was estimated with 1,000 bootstrapping replicates (Tamura et al., 2013). GRAS members in pepper were further categorized into different subfamilies based on well-established classifications in Arabidopsis (Tian et al., 2004).

Protein property and gene structure analysis

With the help of multiple expectation maximization for motif elicitation (MEME, http://meme-suite.org/), conserved motifs of GRAS proteins were scanned with the following parameters: (1) maximum number of motif was 12; (2) optimum motif width was set from 6 to 50 aa (Bailey et al., 2009). These identified motifs were further validated using InterProScan (http://www.ebi.ac.uk/Tools/pfa/iprscan/) (Mulder & Apweiler, 2007). The properties of GRAS proteins were calculated on ExPASy online server (http://web.expasy.org/), such as molecular weight (MW), isoelectric point (pI), instability index and GRAVY value (grand average of hydropathy) (Gasteiger et al., 2003). Based on the relationship of coding sequence and its corresponding genomic DNA sequence, the final exon/intron distribution of each CaGRAS gene was illustrated by GSDS 2.0 (gene structure display server, http://gsds.cbi.pku.edu.cn/) (Hu et al., 2015).

Chromosomal mapping and gene duplication analysis

Physical position of each CaGRAS gene was extracted from pepper genome annotation file, and plotted onto the corresponding chromosome using Mapchart 2.3 (Voorrips, 2002). We renamed each CaGRAS gene according to its ascending chromosomal distribution. Duplicated gene pairs and patterns in pepper were analyzed by using MCScanX and BLASTP (Wang et al., 2012). Tandem duplicated genes were characterized as contiguous homologous genes located in a 100 kb single region or separated by less than five genes, while the whole blocks of genes copying from one chromosome region to another were defined as segmental duplications (Tang et al., 2008). Subsequently, non-synonymous (Ka) and synonymous substitution (Ks) between duplicated CaGRAS gene pairs were calculated by PAL2NAL (http://www.bork.embl.de/pal2nal/) (Suyama, Torrents & Bork, 2006). The microsyntenic map of precise region containing GRAS genes among pepper, tomato and Arabidopsis was created by MCScanX and plotted using Circos (Krzywinski et al., 2009; Wang et al., 2012).

Prediction of CaGRAS protein–protein interaction network

To further clarify the relationships between CaGRASs, a protein–protein interaction network was predicted using their interologs from Arabidopsis. First, specific interolog relationships between Arabidopsis AtGRASs and pepper CaGRASs were mapped from INPARANOID database (http://inparanoid.sbc.su.se/cgi-bin/gene_search.cgi) (Remm, Storm & Sonnhammer, 2001). Then, we retrieved the interaction information among AtGRASs from AraNet database (http://www.functionalnet.org/aranet/) and mapped these attributions to CaGRASs to generate corresponding interaction relationships for pepper (Guo et al., 2015b; Lee et al., 2010). Finally, these interaction networks among CaGRASs were visualized using Cytoscape version 3.4.0 (Shannon et al., 2003).

Expression analysis of CaGRAS genes in different tissues

The public transcriptome data of leaf, stem, root, pericarp and placenta at mature green, breaker, five and 10 days post-breaker, six, 16 and 25 days post-anthesis (PC-MG, PL-MG, PC-B, PL-B, PC-B5, PC-B10, PL-B5, PL-B10, PC-6DPA, PC-16DPA, PC-25DPA, PL-6DPA, PL-16DPA, PL-25DPA) for pepper cv. CM334 have been previously generated (Guo et al., 2015a; Kim et al., 2014). We retrieved the fragments per kilobase per million reads value representing the expression level of each CaGRAS gene and displayed the result using BAR Heatmapper Plus.

Pepper plant preparation and stress treatments

Pepper plants were grown on soil in greenhouse with conditions: 14/10 h photoperiod, 25/20 °C day/night temperature and 60% relative humidity. In this study, pepper seedlings with 6–8 true leaves were randomly divided into five groups, namely control (untreated) and treatment with cold (4±1 °C), salt (300 mM NaCl), drought (400 mM mannitol) and gibberellin solution (20 μM GA). Leaves were sampled at 3 h after the treatment. For each treatment, leaves from five randomly selected seedlings were bulked to form one sample, and six biological replicate samples were immediately frozen in liquid nitrogen and then stored at −80 °C before use.

RNA isolation and qRT-PCR analysis

Total RNA from leaves was extracted using Total RNA kit (BioTeke, Beijing, China) and reversely transcribed into cDNA using M-MLV Reverse Transcriptase (Promega, Madison, WI, USA). Real-time quantitative PCR (qRT-PCR) experiment was done using SYBR GREEN I Master Mix (Applied Biosystems, Waltham, MA, USA) on iCycler iQ™ thermocycle (Bio-Rad, Hercules, CA, USA). Each reaction volume contained 12.5 μl of SYBR GREEN Mix, 1 μl of each primer, 5 μl of 10 × diluted cDNA, and 5.5 μl of nuclease-free water. The reaction program was set as follows: initial polymerase incubation at 95 °C for 10 min, then 40 cycles of 95 °C for 15 s, 60 °C for 45 s. Melting curve analysis was conducted with heating the PCR product from 60 °C to 95 °C for verifying the specificity of the primers. The relative expression levels of CaGRAS genes were calculated based on the comparative Ct method using the 2−ΔΔCt method with the actin1 as an internal reference gene. Primer pairs were designed by Primer Premier 5.0 and checked by NCBI Primer BLAST (Table S3).

Results

Genome-wide identification of GRAS gene family in pepper

We employed two different approaches to identify GRAS genes in pepper genome. Totally, 50 non-redundant CaGRAS genes were found from variety cv. CM334, concurrent with the corresponding genes from cv. Zunla-1 (Table 1). Nearly all these proteins contained one representative GRAS domain (PF03514.11), with the exception of three CaGRASs (CA00g84110, CA01g26680 and CA00g84090) that had more than one such domain. The molecular mass and length of CaGRAS proteins varied greatly, with MWs ranging from 48 to 87 KDa and length from 419 to 801 aa. The average theoretical pI was 6.1, implying that most CaGRAS proteins were weakly acidic. Only CaGRAS21 was stable because of its instability index less than 40, whereas the rest were considered as unstable. All CaGRASs were predicted to be hydrophilic due to the less GRAVY value (<0) of each protein. Most of CaGRAS proteins contained large percentage of aliphatic amino acids, with predicted aliphatic index ranging from 65.74 to 95.76. Interestingly, most of CaGRAS genes (84%) were intronless, while seven members had just one intron. Only one CaGRAS gene had two introns (Fig. 1).

Table 1 Accession members and characteristics of 50 CaGRAS genes in pepper.

ID	Name	Chr	Position (Mb)	Group	Length (aa)	MW (KDa)	pI	Aliphatic index	Instability index	GRVY	Corresponding ID in Zunla-1	
CA01g03260	CaGRAS1	Chr1	5.207267	DLT	679	75.1857	6.2779	83.61	62.53	−0.394837758	Capana01g000318	
CA01g12960	CaGRAS2	Chr1	58.83842	SCL3	473	53.2709	6.4116	90.93	51.59	−0.255932203	Capana00g001336	
CA01g13150	CaGRAS3	Chr1	59.375434	HAM	694	76.1067	5.7836	86.13	60.78	−0.143001443	Capana01g000561	
CA01g23320	CaGRAS4	Chr1	178.490414	Ca_GRAS	590	67.293	4.7921	85.7	43.57	−0.294057725	Capana01g002881	
CA01g23330	CaGRAS5	Chr1	178.494383	Ca_GRAS	565	63.5108	6.2048	84.72	41.97	−0.291489362	Capana01g002882	
CA01g31850	CaGRAS6	Chr1	259.625208	SHR	419	47.7891	5.269	81.17	52.41	−0.468421053	Capana01g003866	
CA02g22690	CaGRAS7	Chr2	157.892927	LAS	588	64.9189	5.3529	81.6	53.65	−0.291111111	Capana02g002687	
CA02g22940	CaGRAS8	Chr2	158.229659	HAM	549	60.6968	5.9537	85.64	42.76	−0.116058394	Capana02g002660	
CA02g25090	CaGRAS9	Chr2	161.98608	SHR	452	50.9813	5.7711	91.24	43.69	−0.155432373	Capana02g002989	
CA02g25280	CaGRAS10	Chr2	162.410539	SHR	529	60.0349	6.23	67.61	50.98	−0.56875	Capana02g002967	
CA02g29990	CaGRAS11	Chr2	169.217032	LISCL	471	53.7333	5.0358	90.32	48.45	−0.280851064	Capana02g003543	
CA03g07840	CaGRAS12	Chr3	23.813921	SHR	563	63.2008	6.2047	80.34	49.19	−0.497330961	Capana03g000095	
CA03g18670	CaGRAS13	Chr3	207.249468	PAT1	642	70.7949	7.1571	82.15	57.55	−0.417316693	Capana03g002179	
CA03g37140	CaGRAS14	Chr3	257.856059	DELLA	551	60.8579	5.0655	80.16	44.9	−0.251636364	Capana03g000088	
CA04g11230	CaGRAS15	Chr4	145.000017	SCR	475	54.4263	6.7004	92.57	41.19	−0.284177215	Capana04g001618	
CA04g11770	CaGRAS16	Chr4	164.396824	HAM	508	56.4831	5.8539	80.45	40.71	−0.281620553	Capana04g002119	
CA04g12860	CaGRAS17	Chr4	178.976341	PAT1	564	62.9078	4.8145	76.41	50.61	−0.387921847	Capana04g001479	
CA05g01670	CaGRAS18	Chr5	2.926557	PAT1	536	59.4974	6.0317	81.87	52.52	−0.234018692	Capana05g000176	
CA05g03110	CaGRAS19	Chr5	7.634233	LISCL	743	83.3188	6.0418	76.77	49.39	−0.473719677	Capana05g000332	
CA05g12700	CaGRAS20	Chr5	182.973215	SHR	567	64.1302	6.5097	79.12	42.35	−0.510600707	Capana05g001029	
CA05g12710	CaGRAS21	Chr5	182.978914	SHR	594	66.9248	6.6408	78.31	37.9	−0.52250423	Capana05g001029	
CA05g17900	CaGRAS22	Chr5	227.253914	PAT1	541	60.2005	5.8782	76.2	47.11	−0.382777778	Capana05g000176	
CA06g00220	CaGRAS23	Chr6	0.170158	Ca_GRAS	562	63.6674	5.6117	88.45	46.66	−0.256327986	Capana00g003286	
CA06g07510	CaGRAS24	Chr6	105.87898	PAT1	608	66.6011	5.5616	85.3	47.88	−0.308731466	Capana00g005111	
CA06g24700	CaGRAS25	Chr6	231.128198	LISCL	751	85.0266	5.1087	74.24	48.17	−0.504266667	Capana06g000410	
CA06g25920	CaGRAS26	Chr6	233.231592	LISCL	681	76.867	6.433	65.74	42.65	−0.593088235	Capana06g000274	
CA07g08560	CaGRAS27	Chr7	124.080011	SCR	440	48.6504	5.1957	95.76	60.13	−0.070615034	Capana07g001083	
CA07g10940	CaGRAS28	Chr7	179.179162	PAT1	542	60.4749	6.7096	82.61	52.9	−0.239001848	Capana07g001257	
CA07g12530	CaGRAS29	Chr7	198.623614	SHR	432	48.7514	5.4183	89.58	43.54	−0.224361949	Capana07g001537	
CA07g14700	CaGRAS30	Chr7	213.163858	HAM	518	59.123	5.069	79.42	51.53	−0.371760155	Capana07g001856	
CA07g18600	CaGRAS31	Chr7	226.087613	PAT1	583	64.5701	6.791	83.83	51.41	−0.300343643	Capana07g002280	
CA07g20170	CaGRAS32	Chr7	228.957285	PAT1	548	61.0729	5.4493	81.3	51.34	−0.27714808	Capana07g002351	
CA07g21550	CaGRAS33	Chr7	231.511151	LAS	449	50.6593	7.9034	93.01	49.39	−0.234821429	Capana07g002493	
CA08g12450	CaGRAS34	Chr8	132.579365	LISCL	688	77.8244	5.7692	74.12	54.14	−0.51516035	Capana08g001582	
CA09g05170	CaGRAS35	Chr9	30.647439	LISCL	763	86.3099	5.6912	78.36	41.38	−0.498031496	Capana09g001814	
CA09g05400	CaGRAS36	Chr9	34.265329	LISCL	748	84.6521	5.219	79.53	44.93	−0.493038822	Capana09g001799	
CA09g13460	CaGRAS37	Chr9	230.053566	PAT1	574	64.3627	5.8404	85.62	42.13	−0.305061082	Capana09g000709	
CA10g04850	CaGRAS38	Chr10	30.048136	LISCL	769	86.7748	6.5228	79.14	44.92	−0.481510417	Capana00g002382	
CA10g10180	CaGRAS39	Chr10	157.782881	SCR	801	87.3531	6.1051	82.91	53.78	−0.34425	Capana10g001031	
CA12g00700	CaGRAS40	Chr12	2.029177	PAT1	533	59.7566	5.6196	80.85	48.1	−0.32537594	Capana12g002864	
CA12g02780	CaGRAS41	Chr12	5.978997	DELLA	597	64.8974	4.7443	78.91	45.53	−0.32147651	Capana05g000798	
CA12g08480	CaGRAS42	Chr12	43.593325	HAM	488	55.5876	5.1055	95.24	43.35	−0.131827515	Capana12g002007	
CA12g21180	CaGRAS43	Chr12	232.987478	Ca_GRAS	518	58.874	5.312	91.8	43.96	−0.27040619	Capana12g000175	
CA12g21800	CaGRAS44	Chr12	233.939464	SCL3	471	52.5634	6.5729	93.57	52.8	−0.142978723	Capana12g000112	
CA00g42950	CaGRAS45	Scaffold1070	0.654105	SCR	519	57.5813	6.7479	83.44	42.65	−0.286293436	Capana00g002482	
CA00g63410	CaGRAS46	Scaffold1392	0.061625	LISCL	787	87.7766	5.7328	83.99	45.41	−0.33129771	Capana05g002208	
CA00g66790	CaGRAS47	Scaffold1455	0.370366	SHR	567	64.1572	6.4197	79.12	41.5	−0.514664311	Capana00g002290	
CA00g67630	CaGRAS48	Scaffold1469	0.384617	HAM	757	82.3972	5.9771	84.13	54.44	−0.218915344	Capana01g003567	
CA00g84090	CaGRAS49	Scaffold1805	0.019698	Ca_GRAS	495	57.4293	5.5379	85.08	43.09	−0.243902439	Capana00g000912	
CA00g84110	CaGRAS50	Scaffold1805	0.106047	Ca_GRAS	445	51.0615	5.3951	88.51	45.76	−0.22454955	Capana00g000912	

Figure 1 Exon-intron structure of CaGRAS genes.

Blue box indicates exon, and black line indicates intron. Y-axis represents the subfamily name of each CaGRAS genes. The lengths of the exons and introns were drawn to scale.

Chromosomal localization and gene duplication analysis of CaGRAS genes

Except for six members (CaGRAS45-50) mapped to the scaffolds, the remaining 44 CaGRAS genes were unevenly distributed across 11 out of 12 pepper chromosomes. Among those anchored members, Chr7 occupied the largest number of GRAS genes (n = 7; 15.22%), followed by Chr1 (n = 6; 13.04%) and three chromosomes (Chr2, Chr5 and Chr12) each having five members. Additionally, four GRAS genes were located on Chr4, while three genes were detected on Chr3, Chr4 and Chr9, respectively. Two GRAS genes were found on Chr8, and only one was on Chr10. Notably, most of CaGRAS genes were gathered at both ends of chromosomes.

Furthermore, we analyzed the duplication events of CaGRAS gene in pepper genome since gene duplication acts importantly on the occurrence of novel functions and gene family expansion. As shown in Fig. 2, two pairs of tandem duplicated genes (CaGRAS4/5 and CaGRAS20/21) located on Chr1 and Chr5, respectively. Additionally, 10 pairs of CaGRAS genes were identified as segmental duplications (Fig. 3). We found that all duplicated gene pairs had Ka/Ks ratios less than 0.5, suggesting these genes experienced strong purifying selection pressure during evolution processes (Table 2). Clearly, segmental duplication played a more prominent role in the expansion of pepper GRAS genes than tandem duplication.

Figure 2 Positions of CaGRAS genes on pepper chromosomes.

Gray shading indicates tandem duplicated region.

Figure 3 Microsynteny analyses of GRAS genes among pepper (Ca), tomato (Sl), and Arabidopsis (At).

Red, yellow and blue lines connecting two chromosomal regions indicate syntenic regions between pepper and tomato, pepper and Arabidopsis, tomato and Arabidopsis chromosomes, respectively. Black lines denote segmental duplicated GRAS genes on the pepper chromosome.

Table 2 Calculation of Ka and Ks ratios of 12 duplicated CaGRAS gene pairs.

Gene pairs	Ka	Ks	Ka/Ks	Duplication Type	Selection Type	
CaGRAS4 vs. CaGRAS5	0.5433	2.9882	0.1818	Tandem	Purifying	
CaGRAS20 vs. CaGRAS21	0.3469	3.0958	0.1121	Tandem	Purifying	
CaGRAS13 vs. CaGRAS24	0.2277	1.2418	0.1834	Segmental	Purifying	
CaGRAS18 vs. CaGRAS22	0.1727	0.6141	0.2813	Segmental	Purifying	
CaGRAS18 vs. CaGRAS32	0.4234	2.6272	0.1611	Segmental	Purifying	
CaGRAS18 vs. CaGRAS40	0.1706	0.6532	0.2612	Segmental	Purifying	
CaGRAS22 vs. CaGRAS40	0.1332	0.6816	0.1955	Segmental	Purifying	
CaGRAS22 vs. CaGRAS32	0.3849	3.4016	0.1132	Segmental	Purifying	
CaGRAS23 vs. CaGRAS43	0.6308	11.5851	0.0545	Segmental	Purifying	
CaGRAS25 vs. CaGRAS35	0.4549	3.1434	0.1447	Segmental	Purifying	
CaGRAS28 vs. CaGRAS32	0.2969	3.0958	0.0959	Segmental	Purifying	
CaGRAS32 vs. CaGRAS40	0.4094	3.8387	0.1066	Segmental	Purifying	
Note:

Ka indicates nonsynonymous substitution rate, and Ks indicates synonymous substitution rate.

In order to understand the phylogenesis of GRAS gene family, microsynteny analysis was employed for the precise region containing GRAS genes in pepper, tomato and Arabidopsis. (Fig. 3). A total of 37, 15 and seven orthologous gene pairs were identified in the cross of pepper and tomato, pepper and Arabidopsis, tomato and Arabidopsis, respectively. Several such regions were also found between different chromosomes in pepper. These data indicate that GRAS gene family is highly conserved, and pepper CaGRAS genes are more closely to those of tomato than that in Arabidopsis. The GRAS genes with microsynteny may evolve from the same ancestor.

Phylogenetic analysis, classification and functional characterization of CaGRAS family

To uncover the evolutionary relationships among CaGRAS proteins and their classifications, we performed a phylogenetic analysis using 189 full-length GRAS proteins (32 from Arabidopsis, 56 from rice, 51 from tomato and 50 from pepper). An unrooted phylogenetic tree was constructed (Fig. 4), demonstrating that all of these GRAS proteins could be classified into eleven distinct subfamilies based on clade support values and classification from Arabidopsis and rice. They were termed as DELLA, PAT1, SCL3, SHR, SCR, LISCL, HAM, LAS, DLT, Ca_GRAS and Os4, respectively. Of these, nine were named specifically according to the findings of previous study (Tian et al., 2004). The remaining two were named by the members of species origin. For example, the subfamily Ca_GRAS consisted of six GRAS members from tomato and six GRAS proteins from pepper, indicating that it might be a Solanaceae-specific group. Os4, a rice-specific subfamily, only contained eleven GRAS members from rice.

Figure 4 Phylogenetic analyses of GRAS proteins from pepper, tomato, rice and Arabidopsis.

The phylogenetic tree was constructed using neighbor-joining (NJ) method by MEGA6.0. Subfamilies were indicated by different colors.

Generally, the genes clustered into a group tend to possess similar function and structure. Therefore, we could predict the potential function of CaGRAS members based on Arabidopsis or rice homologues in the same branch. For example, within the PAT1 subfamily, AtPAT1 and AtSCL13 were previously shown to be involved in phyA and phyB signaling pathway, respectively. Hence, we inferred that CaGRAS28 and CaGRAS18 in the same subfamily may also play a significant role in phytochrome signal transduction. DELLA subfamily included two CaGRAS members (CsGRAS14 and CaGRAS41), five AtGRAS and six OsGRAS. All these members contain the complete DELLA and TVHYNPS motifs (Fig. 5). Previous studies reported that DELLA proteins mainly regulate GA signal transduction pathway (Zhang et al., 2011) implying that CaGRAS14 and CaGRAS41 may have the similar role. The SCL3 subfamily consisted of two CaGRAS members (CaGRAS2 and CaGRAS44) and one AtGRAS (AtSCL3). This subfamily may mediate GA homeostasis through integrating other signals, because AtSCL3 was found to regulate root cell elongation by integrating multiple signals in Arabidopsis (Zhang et al., 2011). For subfamilies SHR and SCR, AtSHR and AtSCR were detected to function importantly in maintaining stem cell and root meristem. It is reasonable to predict that those pepper GRAS homologs in these two subfamilies may possess the similar functions (Di Laurenzio et al., 1996). LISCL subfamily consisted of nine CaGRAS, six AtGRAS and three OsGRAS members, and their biological roles are mostly unknown although a homolog member (LISCL) from Lilium longiflorum was proven to play an important regulatory role during microsporogenesis (Morohashi et al., 2003). The first HAM gene in the HAM subfamily was isolated from petunia and proved to promote shoot indeterminacy (Stuurman, Jaggi & Kuhlemeier, 2002). CaGRAS3 in the HAM subfamily was also demonstrated to be involved in shoot apical meristem organization and axillary meristem development (David-Schwartz et al., 2013). The LAS subfamily comprised two members from pepper, three from rice and three from Arabidopsis. AtLAS proteins in this subfamily mainly function to regulate and promote the initiation of axillary meristems (Liang et al., 2014). The DLT subfamily, the smallest group, contained six members (one from pepper, one from Arabidopsis, two from rice and two from tomato). The members of this group have been previously shown to participate in brassinosteroid signal pathway responsible for the plant height (Tong et al., 2009). For the Ca_GRAS subfamily having six CaGRAS and six SlGRAS members, no Arabidopsis and rice GRAS homolog was grouped into this subfamily, indicating that these genes may be Solanaceae-specific. The function of this subfamily awaits further exploration.

Figure 5 Distribution of conserved motifs in CaGRAS proteins.

(A) The phylogenetic tree and their classification were depicted using the neighbor-joining method in MEGA 6.0. (B) Motif distribution in each GRAS sequence. Motif 10 and 4 is in LHRI domain at N-terminus, followed by Motif 7 and 1 in VHIID domain, Motif 6 and 8 in LHRII domain, Motif 9, 3 and 11 in PFYRE domain, and Motif 2 and 5 in SAW domain at C-terminus.

To investigate the common feature of pepper GRAS proteins in more detail, we used MEME suite to identify their conserved motifs and sequence logos. A total of 11 conserved motifs (named Motif 1–11) were identified, with more motifs locating at C-terminus than at N-terminus. Moreover, the motifs from the same subfamily nearly hold the similar patterns (Fig. 5). We then matched up the motifs with corresponding GRAS domain. It was found that Motif 10 and 4 is in LHRI domain at N-terminus, followed by Motif 7 and 1 in VHIID domain, Motif 6 and 8 in LHRII domain, Motif 9, 3 and 11 in PFYRE domain, and Motif 2 and 5 in SAW domain at C-terminus (Fig. 5). Of the 10 subfamilies of CaGRAS, members from PAT1 and LISCL subfamilies all contained the 11 conserved motifs identified.

Prediction of CaGRAS protein–protein interaction network

Due to unavailable reference for pepper interactome data, we predicted the protein–protein interaction relationships of CaGRAS members based on the interologs from Arabidopsis. We only obtained the interaction information for 19 CaGRAS proteins, and generated a complex interaction network using these proteins (Fig. 6). In general, the members from the SCL3 subfamily (CaGRAS2 and CaGRAS44) owned more interaction partners than others. These were consistent with their working mechanisms, considering the fact that AtSCL3 protein could regulate GA homeostasis by integrating other signal pathway, although such a relationship needs to be confirmed (Zhang et al., 2011). CaGRAS33, a member of LAS subfamily, directly interacted with nine CaGRAS members, while CaGRAS7 from the same subfamily only had three interaction partners. Surprisingly, no interaction partner was detected for CaGRAS proteins from DELLA and DLT subfamily. Our interaction networks may provide important clues for understanding the functions of unknown proteins.

Figure 6 The interaction network of CaGRAS proteins in pepper according to interologs from Arabidopsis.

Expression analysis of CaGRAS genes in various tissues and fruit developmental stages

We used online available transcriptome data of three tissues (leaf, stem and root) and seven developmental stages of pericarp and placenta (mature green, breaker, five and 10 days post-breaker, six, 16, 25 days post-anthesis) to investigate the expression patterns of pepper GRAS genes (Fig. 7). The RPKM value for each of those CaGRAS genes was listed in Table S2. The transcripts for the other 12 CaGRAS genes were not detected in any tissues (RPKM < 0.001), which may be the result of pseudogenes. Generally, 25 CaGRAS genes were detected to express in all tissues, with only five members (CaGRAS8, CaGRAS16, CaGRAS29, CaGRAS38 and CaGRAS48) showing high expression levels (PPKM > 10). A number of CaGRAS genes exhibited a certain degree of tissue specificity. For example, CaGRAS18 and CaGRAS27 were only expressed in pepper pericarp. CaGRAS35 and CaGRAS43 were highly expressed in leaf while the transcripts of CaGRAS30 and CaGRAS34 largely accumulated in stem rather than in other tissues. Tissue-specific expression of these genes showed that they may highly participate in the corresponding tissue development. CaGRAS28 homologous with AtPAT1 showed high expression level in leaves, which is in line with AtPAT1 function as a positive regulator in phyA signal pathway (Bolle, Koncz & Chua, 2000). Several CaGRAS genes exhibited constitutive expression levels at most stages of pericarp development. For example, CaGRAS7 and CaGRAS42 displayed a relatively higher expression at green fruit stage (PC_6DPA and PC_16DPA), and then decreased gradually towards fruit ripening. This expression pattern implied that CaGRAS7 and CaGRAS42 may function importantly in the early fruit development. In addition, the similar expression patterns were often detected for gene pairs from duplication event, but not for all such genes. For instance, in the CaGRAS18/40 duplicated region, CaGRAS40 was highly expressed, whereas the other showed the opposite expression pattern. These differences implied that duplicated GRAS gene pairs may have diverged evolutionary outcomes.

Figure 7 Heatmap and hierarchical clustering of CaGRAS genes in leaf, stem, root and mature green (MG), breaker (B), five and 10 days post-breaker (B5, B10), six, 16, 25 days post-anthesis (6DPA, 16DPA, 25DPA) of pericarp (PC) and placenta (PL).

The expression values were calculated by RPKM measure and then were log2 transformed before generating heat maps.

Response of CaGRAS genes to different stress treatments

In order to elucidate the functions of CaGRAS genes responsive to GA stimuli, qRT-PCR was performed to examine the expression of such genes in seedling leaves after treatment with GA. In this study, 14 CaGRAS genes showed obvious changes in response to GA treatment (Fig. 8). Of them, the most upregulated gene was CaGRAS37, while the most downregulated gene was CaGRAS10. To broaden our knowledge regarding how these genes are affected by GA, we conducted a comprehensive analysis on cis-elements in the promoter regions of such 14 CaGRAS genes using PlantCARE (Lescot et al., 2002). Additionally, 12 CaGRAS genes were detected to contain at least one GA responsive element (GARE) in their promoter sequences, again confirming the function of these genes in mediating GA signal pathway in pepper (Table S4).

Figure 8 Differential expression analyses of 21 GRAS genes under GA, drought, salt and cold treatment in pepper seedlings.

The color scale represents log2 expression values.

We further examined the expression levels of CaGRAS genes under abiotic stresses, including salt, drought and cold treatments. Compared to the control group, the expression of 12 CaGRAS genes were highly affected by these treatments, indicating that those genes may have diverse functions involving in plant responses to abiotic stresses. The downregulated expression was detected for six, two and four CaGRAS genes, respectively, under cold, drought and salt stresses. Furthermore, we found the upregulated genes exhibit a group-specific expression. For example, the expression of CaGRAS genes from the DELLA subfamily was significantly induced under cold stress. The genes in the SCL3 subfamily was highly upregulated under drought stress, and the genes in the PAT1 subfamily were highly induced by GA and other four stress treatments. Therefore, it is possible that different CaGRAS members function in different stress responses.

Discussion

With the rapid development of bioinformatics, information stored in genome sequence is increasingly to become the target to explore the mechanism underlying plant growth and development. Recent studies in a number of higher plants by comparative genomics showed that GRAS transcription factors play significant roles in multiple biological processes (Huang et al., 2015; Lee et al., 2008; Wu et al., 2015; Xu et al., 2016). However, limited knowledge was available for GRAS genes in pepper. In this study, we conducted a systematic analysis on this important transcription factor family in pepper, including genome-wide identification of CaGRAS members, chromosomal localization, intron-exon structure, physical-chemical features, phylogenetic analysis, duplication events, microsyntenic mapping and expression profiles in various pepper tissues as well as their responses to different stresses.

A total of 50 CaGRAS genes were obtained from 34,903 protein-coding genes in pepper genome. The number of CaGRAS genes is actually more than that in Arabidopsis (32), P. mume (45), castor bean (46) (Lee et al., 2008; Lu et al., 2015; Xu et al., 2016), comparable to cabbage (48) and tomato (53) (Huang et al., 2015; Song et al., 2014), but less than those in rice (60) and Populus (106) (Tian et al., 2004). The variation of GRAS gene number might be related to gene duplication events or genome size. This study detected two pairs of tandem duplicated CaGRAS genes and 10 pairs of segmental duplicated CaGRAS genes. However, 15 SlGRAS members were identified as tandem duplications in tomato. It looks like that segmental duplication contribute more to pepper GRAS expansion than tandem duplication whereas tandem duplication may be major player in this regard for tomato. Moreover, pepper genome size (3.48 Gb) is about fourfold larger than tomato genome (900 Mb), indicating that expansion mechanisms of GRAS genes are different among lineages.

All 50 CaGRAS proteins were classified into 10 subfamilies according to their conserved domains and sequence homology in Arabidopsis and rice (Tian et al., 2004). Notably, we observed a Solanaceae-specific subfamily (Ca_GRAS) contained the members from pepper and tomato but no Arabidopsis and rice GRAS homolog, whereas a rice-specific subfamily (Os4) was not detected in Arabidopsis, tomato and pepper. In agreement of this, the species-specific GRAS subfamily also widely existed in other plant species, such as the Rc_GRAS subfamily in castor bean (Xu et al., 2016) and the Pt20 subfamily in Populus (Liu & Widmer, 2014). These species-specific GRAS genes might be lost from other plants or become highly specialized during evolution.

The categorization of CaGRAS family was further supported by analysis of conserved motifs in those pepper proteins. Conserved motifs were found within the GRAS domain regions which might function importantly. Although conserved motifs were identical among all CaGRAS proteins, a number of differences in chemical–physical characteristics were also detected for CaGRAS members. These differences may due to the amino acid discrepancies in the non-conserved regions of CaGRAS members, implying that different CaGRAS proteins may act different functions in their own microenvironments (Huang et al., 2015).

Another important finding is that most CaGRAS genes (84%) contain just one exon. The high percentage of such intronless GRAS genes is detected as 67.6%, 54.7%, 82.2% and 83.3% in Arabidopsis, Populus, P. mume and Chinese cabbage (Lee et al., 2008; Lu et al., 2015; Song et al., 2014; Tian et al., 2004), respectively, evidencing again that the GRAS proteins were highly conserved among those plant species. Besides GRAS genes, intronless genes were also enriched among some other gene families, such as SAUR genes, F-box gene families and DEAD box helicases (Aubourg, Kreis & Lecharny, 1999; Jain et al., 2007; Jain, Tyagi & Khurana, 2006). Given the fact that intronless genes are archetypical in prokaryotic genomes, the recent work by Zhang, Iyer & Aravind (2012) showed that the origin of plant GRAS genes is derived from the prokaryotic genomes by horizontal gene transfer, followed by duplication events in evolutionary history. This may explain the formation of substantial intronless GRAS genes in pepper genome.

Generally, an intrinsically disordered region (IDR) in an intrinsically disordered protein (IDP) allows protein to recognize and interact with various partners, which are crucial for molecular function. Bioinformatics analysis showed that GRAS protein is a kind of IDP (Sun et al., 2013). One of a typical IDR in GRAS protein is its highly variable N-terminus, which possess short interaction-prone segments and molecular recognition features responsible for recognizing and binding the specific partner of GRAS proteins. Here, pepper GRAS proteins were found to contain a highly variable N-terminal region, which is consistent with the notion that N-terminus of GRAS proteins were intrinsically disordered, contributing to the functional divergence of CaGRAS proteins.

For functional characterization of those CaGRAS genes, we performed an extensive analysis for their expression profiles in different tissues and stress conditions, particularly for those in pepper-specific subfamily without function information deduced from model plant Arabidopsis or rice. Our data showed that CaGRAS4 may be a pseudogene because of no expression level detected in any tissues. CaGRAS5 might be involved in pericarp and placenta development, showing a relatively high abundance during all consecutive development stages. On the whole, the expression profiles of CaGRAS genes varied greatly not only among different tissues, but members from the same subfamily. Likely, such a great expression variation was also observed for GRAS genes in Populus and P. mume (Huang et al., 2015). These results indicated that GRAS genes may have experienced neo-functionalization or sub-functionalization in many higher plants. The RPKM values of twelve CaGRAS genes from seven subfamilies (DELLA, PAT1, SHR, SCR, LISCL, LAS and Ca_GRAS) were not detected in any tissues, suggesting these genes may lose their functions during evolution. By contrast, higher expression levels of CaGRAS genes in several organs signified their important roles. For example, CaGRAS29 from the SHR subfamily was highly transcribed in root tissue, which is consistent with the function of its homologous AtSHR responsible for root development (Cui et al., 2007). CaGRAS41 from the DELLA subfamily expressed in all tissues played critical roles in controlling a variety of signal hubs, whereas no expression of CaGRAS2 from the same subfamily was detected in any tissues. It seems that functional diversification is occurred for the two CaGRAS genes from the DELLA subfamily. Overall, the current expression data obtained for CaGRAS genes in different tissues lay a foundation for further functional analysis of pepper GRAS members.

In general, hormones could regulate plant growth and development via the modulation of the related gene expression. GA is found to play important roles in many aspects of plant development such as organ elongation, germination and flowering time. It has been reported that expression of GRAS genes in tomato showed dose-dependent response to GA (Huang et al., 2017). Our results demonstrated that the majority of CaGRAS genes detected here displayed dramatic changes after GA treatment. The promoters of these CaGRAS genes contained at least one GARE, implying that a set of CaGRAS proteins could regulate plant adaptability to adversity through a complex regulatory network. Additionally, previous studies revealed that GRAS genes could affect plant responses to abiotic stresses. For example, BnLAS and PeSCL7, GRAS members from Brassica napus and poplar, were identified as the good targets for engineering to increase plant drought and salt tolerance (Ma et al., 2010; Yang et al., 2011). Combined analysis of all qPCR results revealed that several pepper GRAS genes were associated with the above three stress responses (cold, salt and drought), showing the cross-talking of GRAS genes in regulation of plant responses against various adversity. Notably, we found that CaGRAS members belonging to PAT1 group exhibit the similar expression patterns when stressed by GA and other abiotic treatments. Consistently, OsGRAS genes from rice PAT1 group were also reported to be involved in GA and stress responses. All these indicate that some GRAS genes may specifically coordinate plant responses to multiple stresses.

Conclusion

In this study, 50 CaGRAS members were characterized from pepper genome, and classified into 10 subfamilies based on phylogenetic relationships. Duplication event particularly segmental duplication was identified as the main driving force to GRAS gene expansion in pepper. Interaction network and expression profiles among CaGRAS genes were examined, illustrating important roles of CaGRAS proteins in regulating GA and abiotic stress responses. Taken together, our study is the first comprehensive characterization of GRAS genes in pepper. All these data provide the foundation to elucidate the GRAS-mediated molecular mechanism underlying plant growth and development as well as stress biology, showing that GRAS members could be selected as the targets for genetic improvement of stress tolerance in pepper and other related plants.

Supplemental Information

Supplemental Information 1 Gene IDs from rice, tomato and Arabodipsis.

Click here for additional data file.

Supplemental Information 2 The RPKM value of each gene in different tissues and development stages.

Click here for additional data file.

Supplemental Information 3 Primers’ sequences used for studies of CaGRAS expression.

Click here for additional data file.

Supplemental Information 4 Cis-elements analysis of pepper GRAS genes.

Click here for additional data file.

Additional Information and Declarations

Competing Interests

Author Contributions

Data Availability

The authors declare that they have no competing interests.

Baoling Liu conceived and designed the experiments, performed the experiments, contributed reagents/materials/analysis tools, prepared figures and/or tables, authored or reviewed drafts of the paper, approved the final draft.

Yan Sun performed the experiments, analyzed the data, prepared figures and/or tables, authored or reviewed drafts of the paper, approved the final draft.

Jinai Xue performed the experiments, analyzed the data, prepared figures and/or tables, authored or reviewed drafts of the paper, approved the final draft.

Xiaoyun Jia conceived and designed the experiments, authored or reviewed drafts of the paper, approved the final draft.

Runzhi Li conceived and designed the experiments, contributed reagents/materials/analysis tools, authored or reviewed drafts of the paper, approved the final draft.

The following information was supplied regarding data availability:

The research in this article did not generate any raw data.

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
