# Peer review of "Genome-wide characterization and expression analysis of GRAS gene family in pepper (Capsicum annuum L.)"

_PeerJ, doi:10.7717/peerj.4796_

## Round 0.1 · original submission · Major Revisions

Please address all reviewers comments.

Reviewer 1 ·

Basic reporting

This article deals with the identification and characterization of GRAS gene family in chili pepper (Capsicum annuum L.). In general, the article is clear, with sufficient field background cited literature, well-structured and organized, with adequate support materials (figures, tables and additional tables), and with interesting and relevant results about this gene family. However, some writing minor mistakes were detected and marked in the text and in References.

Experimental design

Experimental design of this study is considered adequate, with a well defined question and appropriate experimental approach to answer it. This research is considered as well performed and with high technical standard. However, a just one possible weak part is that regarding the qRT-PCR expression analysis using actin as reference gene, because very often three reference genes are included for this kind of analysis due to some large variation of expression levels of them, as it has been particularly reported for actin gene in chili pepper considered as the least stable reference gene (see Wan et al., Biochem. Biophys. Res. Commun. 416:24-30 (2011).

Validity of the findings

In general, the overall analyses and results are very interesting and will be certainly of great value for the knowledge of GRAS gene family in chili pepper, which will be to encourage deeply research on the diverse functions of these genes.

Additional comments

Please attend all the marked corrections and observations indicated in the text of the attached file, following carefully the Instructions to Authors of this journal.

Annotated reviews are not available for download in order to protect the identity of reviewers who chose to remain anonymous.

Reviewer 2 ·

Basic reporting

Overall, the article is well written and well presented. The language at some sentences can be improved for clarity e.g. lines 9, 145, 170, 179, 271, 280, 359.

Experimental design

The article presents the analyses of GRAS gene family in pepper. Most of the manuscript is descriptive with different bioinformatic analyses of the GRAS genes.

Major comment:
- The rational of analyzing GRAS in pepper is unclear. GRAS family has been characterized in 30 plant species (lines 39-41), and authors do not discuss what's different in pepper GRAS genes compared to their orthologs in other species. I recommend that authors strengthen rational of their analyses and discuss the new insights and knowledge-gap that is achieved by their manuscript.

Minor comments:
- Abstract line 19: "...play vital roles in..." is too strong a claim. Expression does not imply a vital role. It should be "...may be implicated in...", or something to that effect.

- Line 133-135: How was the RPKM values obtained? If it was from the original published study, it should be clearly mentioned.

- Line 187-190: Are some of the GRAS paralogs originated, from whole genome duplications in plants? No evidence is presented for this.

- Line 264-265: Have authors looked for evidence that these genes are pseudogenes?

Validity of the findings

Findings may only be useful for a limited set of researchers.

Reviewer 3 ·

Basic reporting

While the paper conveys the scientific message, it needs grammatical editing. The introduction is not coherent and contains too much information not pertaining to the study.

Experimental design

The authors should clarify how their pipeline is different than the previously published papers. It would be helpful to see novelty rather than repeating template bioinformatics pipeline on different TF families or new plant varieties.

Validity of the findings

Please explain why use (only) these two cultivars of capsicum and using Arabidopsis to compare instead of some other close relative?

I think the authors and the paper would reach a broader audience and benefit more if they can include some more TF families in this study. Especially since most of the work is bioinformatics analysis, I strongly urge expanding the study to include couple other important transcription factors or at the very least include few other species or cultivars of capsicum.

Additional comments

Minor:
How many samples were used for figure 2?

Why do authors cite papers in the results section? They can relate their findings to previous literature in the discussion.

Why did the authors choose the “The length of all hits out of the range from 350 to 820 aa was rejected.”

---

## Round 0.2 · accepted · Accept

Congratulations

All three re-reviewers are in agreement. Your manuscript has been Accepted for publication

# Reviewer 1 ·

Basic reporting

Authors attended the comments and observations made by the reviewers.

Experimental design

Authors attended the comments and observations made by the reviewers.

Validity of the findings

Authors attended the comments and observations made by the reviewers.

Additional comments

Authors attended the comments and observations made by the reviewers.

Reviewer 2 ·

Basic reporting

The manuscript is much improved compared to the first submission.

Experimental design

No comments.

Validity of the findings

The scope is improved with additional analyses and data. Although this will still be only a little advance in the field without much insights.

Additional comments

The authors have satisfactorily answered my comments.

Reviewer 3 ·

Basic reporting

N/A

Experimental design

N/A

Validity of the findings

N/A